# Label Noise-Robust Learning using a Confidence-Based Sieving Strategy

## Abstract

In learning tasks with label noise, boosting model robustness against overfitting is a pivotal challenge because the model eventually memorizes labels including the noisy ones. Identifying the samples with corrupted labels and preventing the model from learning them is a promising approach to address this challenge. Per-sample training loss is a previously studied metric that considers samples with small loss as *clean* samples on which the model should be trained. In this work, we first demonstrate that this small-loss trick is not efficient by itself. Then, we propose a novel discriminator metric called *confidence error* and a *sieving* strategy called CONFES to effectively differentiate between the clean and noisy samples. We experimentally illustrate the superior performance of our proposed approach compared to recent studies on various settings such as synthetic and real-world label noise. Moreover, we show CONFES can be combined with other approaches such as Co-teaching and DivideMix to further improve the model performance.

## 1 Introduction

The superior performance of deep neural networks (DNNs) in numerous application domains, ranging from medical diagnosis De Fauw et al. (2018); Liu et al. (2019) to autonomous driving Grigorescu et al. (2020) mainly relies on the availability of large-scale and high quality data Sabour et al. (2017); Marcus (2018). Supervised machine learning in particular requires correctly annotated datasets to train highly accurate DNNs. However, such datasets are rarely available in practice due to labeling errors (leading to *label noise*) stemming from high uncertainty Beyer et al. (2020) or lack of expertise Peterson et al. (2019). In medical applications for instance, there might be a disagreement between the labels assigned by radiology experts and those from the corresponding medical reports Majkowska et al. (2020); Bernhardt et al. (2022), yielding datasets with noisy labels.

In real-world scenarios, a corrupted label assigned to a sample depends on the feature values and the true label of the sample, which is known as *instance-dependent noise* Liu (2021); Zhang et al. (2021b). There are also other types of label noise in the literature including *symmetric noise*, where a sample is allocated a random label or *pairflip noise*, in which the label of a sample is flipped into the adjacent label Patrini et al. (2017); Xia et al. (2020); Bai et al. (2021). Instance-dependent noise is more challenging than the other label noise types Yao et al. (2020); Berthon et al. (2021). Training DNNs in the presence of label noise can lead to memorization of noisy labels, and consequently, reduction in model generalizability Zhang et al. (2021a); Chen et al. (2021b). Hence, it is indispensable to design and develop robust learning algorithms that are able to alleviate the adverse impact of noisy labels during training. Throughout this paper, we will refer to these methods as *label noise learning* methods.

Some of the existing studies Patrini et al. (2017); Berthon et al. (2021); Yao et al. (2021b); Xia et al. (2019); Yao et al. (2020) model the noise distribution as a transition matrix, encapsulating the probability of clean labels being flipped into noisy ones, and leverage loss correction to attenuate the effect of the noisy samples. Other studies Cheng et al. (2020); Xia et al. (2021); Wei et al. (2020) learn the clean label distribution and capitalize on regularization or selection of reliable samples to cope with the noisy labels. Although the former line of work theoretically guarantees the consistency between the classifiers learned with and without

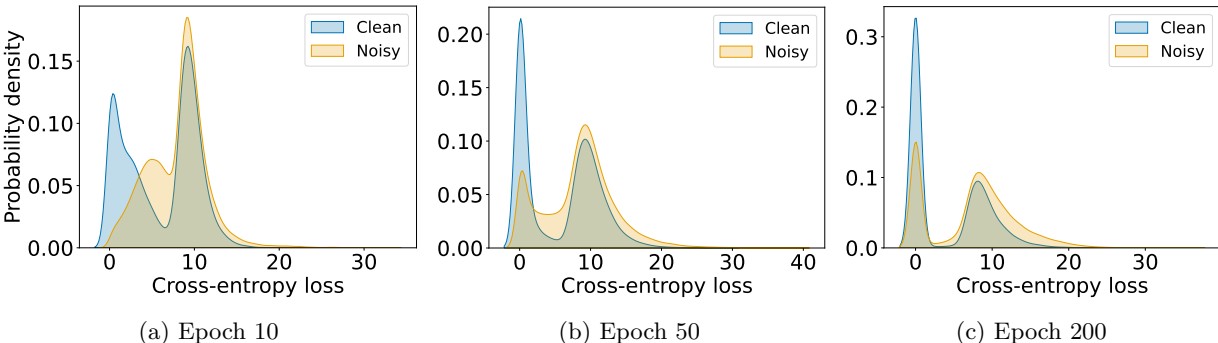

(a) Epoch 10             (b) Epoch 50             (c) Epoch 200

Figure 1: **Distributions of loss values** for clean and noisy labels are relatively similar, and consequently, loss value is a less-efficient metric to distinguish between the clean labels and noisy ones. Experiments are performed using PreAct-ResNet18 trained on CIFAR-100 with instance-dependent label noise of level 60%.

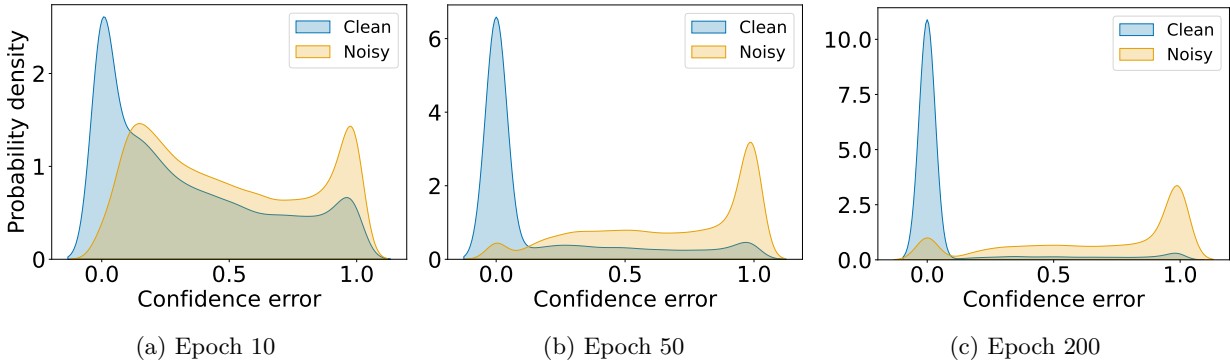

(a) Epoch 10             (b) Epoch 50             (c) Epoch 200

Figure 2: **Distributions of confidence error values** for clean and noisy samples progressively diverge from each other as the training process continues. This indicates that confidence error is a more effective metric than loss value to differentiate the clean samples from noisy ones. Experiments are conducted using PreAct-ResNet18 and CIFAR-100 with noise level of 60%.

noisy samples, its efficiency strongly depends on the accurate estimation of the transition matrix, which is a non-trivial task Xia et al. (2019); Bernhardt et al. (2022); Cheng et al. (2020).

A main challenge in the latter line of work, also known as sample selection (sieving), is to find a reliable criterion (or metric), which can efficiently differentiate between clean and noisy samples. The majority of the previous studies Jiang et al. (2018); Han et al. (2018); Yu et al. (2019) employ the *loss value* to this end, where the samples with small loss values are considered to likely be clean ones (*small-loss trick*). However, our observations (Fig 1) illustrate the distributions of the loss values for clean and noisy samples overlap widely, implying that a lot of noisy samples have small loss values and vice versa. Consequently, loss value is not an effective metric to distinguish between clean and noisy samples.

**Contributions.** We propose a novel metric called *confidence error* to more efficiently discriminate between the clean and noisy labels. The confidence error metric is defined as the difference between the softmax outputs/logits of the predicted and original label of a sample. Our observations (Fig 1) indicate that there exist a clear correlation between the confidence error value and the probability of being clean. That is, a sample with lower confidence error has much higher probability to be a clean sample than a noisy one.

Next, we integrate the confidence error criterion into a novel learning algorithm called *CONFidence Error Sieving* (CONFES) to robustly train DNNs in the instance-dependent, symmetric, and pairflip label noise settings. The CONFES algorithm computes the confidence error associated with training samples at the beginning of each epoch and only incorporates a subset of training samples with lower confidence error values

during training (i.e. likely clean samples). We draw a performance comparison between CONFES and its competitors using typical benchmark datasets for label noise learning including CIFAR-10/100 Krizhevsky et al. (2009) and Clothing1M Xiao et al. (2015).

In summary, we make the following contributions:

- We demonstrate that loss value by itself is not an effective metric to discriminate between the clean and noisy samples.

- To address this shortcoming, we introduce the confidence error as a novel alternative metric and illustrate that it can efficiently differentiate clean samples from noisy ones.

- We propose the CONFES learning algorithm, which leverages the confidence error as a core building block to effectively *sieve* the training samples in an *online fashion* during training.

- Through extensive experiments, we show that CONFES outperforms the state-of-the-art learning algorithms of comparison, in terms of accuracy in symmetric, pairflip, and instance-dependent label noise scenarios.

- We combine CONFES with other learning algorithms including CoTeaching, JoCor, and DivideMix and illustrate the combination provides further improved accuracy.

## 2 Related Work

Overcoming the memorization of noisy labels plays a crucial role in label noise learning and improves model generalization by making the training process more robust to label noise Song et al. (2022); Zhang et al. (2021a); Natarajan et al. (2013); Arpit et al. (2017). The research community mainly tackled the memorization problem by adjusting the loss function Yao et al. (2020); Xia et al. (2019); Yao et al. (2021b), using implicit/explicit regularization techniques Xia et al. (2021); Liu et al. (2020); Zhang et al. (2018); Chen et al. (2021a); Song et al. (2019a), or refining the training data and performing sample sieving (also called sample selection) Han et al. (2018); Jiang et al. (2018); Yu et al. (2019); Cheng et al. (2020); Li et al. (2020); Yao et al. (2021a); Malach & Shalev-Shwartz (2017); Kim et al. (2021); Pleiss et al. (2020); Hu et al. (2021).

Adjusting the loss function according to the noise transition probabilities is an effective method for decreasing the adverse impact of noisy samples during the training but comes at the cost of accurate estimation of the transition matrix Patrini et al. (2017). Previous studies Yao et al. (2020); Xia et al. (2019); Yao et al. (2021b) have paved the way for this non-trivial estimation in different ways. For instance, T-Revision Xia et al. (2019) estimates the transition matrix without requiring anchor points (the data points whose associated class is known almost surely), which play an important role in learning the transition matrix effectively. Dual-T Yao et al. (2020) first divides the transition matrix into two matrices that are easier to estimate, and then aggregates their outputs for more accurate estimation of the original transition matrix.

A different line of work Liu et al. (2020); Xia et al. (2021); Zhang et al. (2018); Chen et al. (2021a) improves model generalization by introducing regularization effects suitable for learning with noisy labels. The regularization effect may be injected implicitly using methods such as data augmentation Zhang et al. (2018) and inducing stochasticity Chen et al. (2021a). For example, Mixup Zhang et al. (2018) augments the training data using a convex combination of a pair of examples and their labels which encourages the model to learn a simple interpolation between the samples. SLN Chen et al. (2021a) introduces stochastic label noise to help the optimizer to skip sharp minima in the optimization landscape.

Although conventional (implicit) regularization techniques such as dropout Srivastava et al. (2014) and data augmentation have been proven effective in alleviating overfitting and improving generalization, they are insufficient to tackle the label noise challenge Song et al. (2022). ELR (Early-Learning Regularization) Liu et al. (2020) is an explicit regularization approach based on the observation that at the beginning of training, there is an early-learning phase in which the model learns the clean samples without overfitting the noisy ones. Given that, ELR adds a regularization term to the Cross-Entropy (CE) loss, leading the model output towards its own (correct) predictions at the early-learning phase. Similarly, CDR Xia et al. (2021) groups

the model parameters into critical and non-critical in terms of their importance for generalization, and then penalizes the non-critical parameters.

Differentiating the clean samples from the noisy ones (known as sample selection/sieving) and employing them in the training process, is a promising direction to enhance robustness in label noise learning Han et al. (2018). MentorNet Jiang et al. (2018) uses an extra pretrained model (mentor) to help the main model (student) by providing it with small-loss samples. The decoupling algorithm Malach & Shalev-Shwartz (2017) trains two networks simultaneously using the samples on which the models disagree about the predicted label. Co-teaching Han et al. (2018), cross-trains two models such that each of them leverages the samples with small-loss values according to the other model. Co-teaching+ Yu et al. (2019) improves Co-teaching by considering the clean samples as the ones that not only have small loss but also those on which the models disagree. JoCoR Wei et al. (2020) computes a joint-loss to make the outputs of the two models become closer, then it considers the samples with small loss as clean samples.

---

**Algorithm 1:** Confidence error based sieving (CONFES)

---

**Input:** Noisy training dataset $\tilde{D} = \{(x_i, \tilde{y}_i)\}_{i=1}^n$, model $\mathcal{F}_\theta$, number of training epochs $T$, initial sieving threshold $\alpha$, number of warm-up epochs $T_w$, batch size $M$

**Output:** Trained model $\mathcal{F}_\theta$

1 **for** $i = 0, ..., T$ **do**
2     $\alpha_i = \text{Max}(\alpha - i \cdot \frac{\alpha}{T_w}, 0)$ /* Set sieving threshold                    */
3     $O^i \leftarrow \mathcal{F}_\theta^i(\tilde{D})$ /* Feed the dataset to model                    */
4     $\mathcal{C}^i \leftarrow \mathcal{P}(O^i)$ /* Calculate confidence on labels              */
5     $E_C(s) = C^{(\hat{y}_i)} - C^{(\tilde{y}_i)}$ /* Compute confidence error             */
6     $D_c = \{s \mid E_C(s) \leq \alpha_i\}$ /* Sieve clean samples                  */
7     $D^{'} = D_c \oplus (\{(\text{augment}(x_i), \tilde{y}_i)\} \mid (x_i, y_i) \subseteq D_c)$. /* Build new dataset(clean⊕augmented)      */
8     **for** *mini-batch* $\beta = \{(x_i, \tilde{y}_i)\}_{i=1}^M \in D^{'}$ **do**
        /* Train the model on new dataset                    */
9        Update model $\mathcal{F}_\theta$ on mini batch $\beta$ using Equation equation 1
10 **return** Trained model $\mathcal{F}_\theta$

---

## 3 CONFES: CONFidence Error based Sieving

We assume a classification task on a training dataset $D = \{(x_i, y_i) \mid x_i \in X, y_i \in Y\}_{i=1}^n$, where $n$ is the number of samples and $X$ and $Y$ are the feature and label (class) space, respectively. The neural network model $\mathcal{F}(X_\beta; \theta) \in \mathbb{R}^{m \times k}$ is a $k$-class classifier with trainable parameters $\theta$ that takes mini-batches $X_\beta$ of size $m$ as input. In real life, a sample might be assigned a wrong label (e.g. due to human error). Consequently, *clean* (that is, label noise-free) training datasets are not always available in practice. Given that, assume $\tilde{Y} = \{\tilde{y}_i\}_{i=1}^n$ and $\tilde{D} = \{(x_i, \tilde{y}_i)\}_{i=1}^n$ indicate the noisy labels and noisy dataset, respectively. The training process is conducted by minimizing the empirical loss (e.g. cross-entropy) using mini-batches of samples from the noisy dataset:

$$\min_\theta \mathcal{L}(\mathcal{F}(X_\beta; \theta); \tilde{Y}_\beta) = \min_\theta \frac{1}{m} \sum_{i=1}^m \mathcal{L}(\mathcal{F}(x_i, \theta), \tilde{y}_i), \tag{1}$$

where $\mathcal{L}$ is the loss function and $(X_\beta, \tilde{Y}_\beta)$ is a mini-batch of samples with size $m$ from the noisy dataset $\tilde{D}$.

In the presence of label noise, the efficiency of the training process mainly depends on the capability of the model to distinguish between clean and noisy labels and to diminish the impact of noisy ones on the training process. The loss value is a commonly used metric to that end, where a sample with lower loss value is considered to be more likely a clean sample than a noisy one Han et al. (2018); Jiang et al. (2018); Yu et al. (2019); Li et al. (2020). We design an experiment to investigate the effectiveness of this consideration: we employ the SGD optimizer and cross-entropy loss function to train PreActResNet18 on CIFAR-100, where 60% of the labels are made noisy using the instance-dependent noise. At the beginning of each epoch, the

model computes the loss value for all training samples, and sorts them in ascending order by loss value. The model considers the first 40% of the samples with lower loss values as clean, and only incorporate them during training. This procedure is repeated for 200 epochs.

Figure 1 visualizes the probability density function, computed using the kernel density estimation method, of cross-entropy loss values. As seen in the figure, the distribution of the loss values for the clean and noisy labels are relatively analogous. Moreover, the corresponding distributions still remain similar as the training process continues. This implies that the loss value by itself is not necessarily an effective metric to discriminate between the clean and noisy samples. Consequently, the prior studies employ different techniques such as loss correction Patrini et al. (2017); Arazo et al. (2019) or relying on an additional model (e.g. co-teaching) Han et al. (2018); Li et al. (2020); Yao et al. (2021a); Jiang et al. (2020) to enhance the robustness of cross-entropy loss to label noise. In this study, we take a different approach and propose an alternative metric called *confidence error* for more efficient sieving of the samples during training.

**Confidence Error**   Consider a sample $s = (x_i, \tilde{y}_i)$ from the noisy dataset $\tilde{D}$. The $k$-class/label classifier $\mathcal{F}(x_i; \theta))$ takes $x_i$ as input and computes the weight value associated with each class as output. Moreover, assume $\mathcal{P}(\cdot)$ is the softmax activation function such that $\mathcal{P}(\mathcal{F}(x_i; \theta)) \in [0, 1]^k$ takes classifier's output and computes the predicted probability for each class. We define the *model confidence* for a given label $l \in \{1, \ldots, k\}$ associated with sample $s$ as the prediction probability assigned to the label:

$$C^{(l)} = \mathcal{P}(\mathcal{F}(x_i; \theta))^{(l)} \tag{2}$$

The class with the maximum probability is considered as the predicted class, i.e. $\hat{y}_i$, for sample $s$:

$$\hat{y}_i = \underset{j \in \{1, \ldots, k\}}{\arg\max} \ \mathbb{P}^{(j)}(\mathcal{F}(x_i; \theta)), \tag{3}$$

where $\mathbb{P}$ indicates probability. The *confidence error* $E_C(s)$ for sample $s$ is defined as the difference between the probability assigned to the predicted label $\hat{y}_i$ and the probability associated with the original label $\tilde{y}_i$:

$$E_C(s) = C^{(\hat{y}_i)} - C^{(\tilde{y}_i)}, \tag{4}$$

where $E_C(s) \in [0, 1]$. In other words, the confidence error states how much the model confidence on the original class is far from the model confidence on the predicted class. The confidence error of zero implies that the original and predicted classes are the same.

We repeat the previous experiment, but with confidence error as the metric to differentiate the clean samples from noisy ones. As shown in Figure 2, the distribution of confidence error values for the clean and noisy samples becomes more and more dissimilar as the training process proceeds. For instance at epoch 50, a sample with high confidence error (e.g. near 1.0) is much more likely to be a noisy sample than clean one. Likewise, a sample with very low confidence error is most probably a clean sample. Given that, we conclude that confidence error is a more efficient metric than loss value to distinguish between the clean and noisy samples.

**CONFES algorithm**   Previous studies Bai et al. (2021); Liu et al. (2020) show that deep neural networks tend to memorize noisy samples, which can have detrimental effect on the model utility. Therefore, it is crucial to detect the noisy samples and alleviate their adverse impact, especially in the early steps of training. The CONFES algorithm takes this into consideration by sieving the training samples using the confidence error metric and *completely excluding* the identified noisy samples during training. CONFES (Algorithm 1) consists of three main steps at each epoch: Sieving samples, building the refined training set, and training the model.

In the sieving step, the confidence error for each training sample is computed using Equation 4; then, the samples whose confidence error is less than or equal to $\alpha_i$ (sieving threshold at epoch $i$) are considered as clean, whereas the remaining samples are assumed to be noisy and excluded from training. CONFES has two hyper-parameters: initial sieving threshold $\alpha$, and the number of warm-up epochs $T_w$, which are used to compute per-epoch sieving threshold $\alpha_i$ at each epoch $i$. In the second step, a new training dataset is

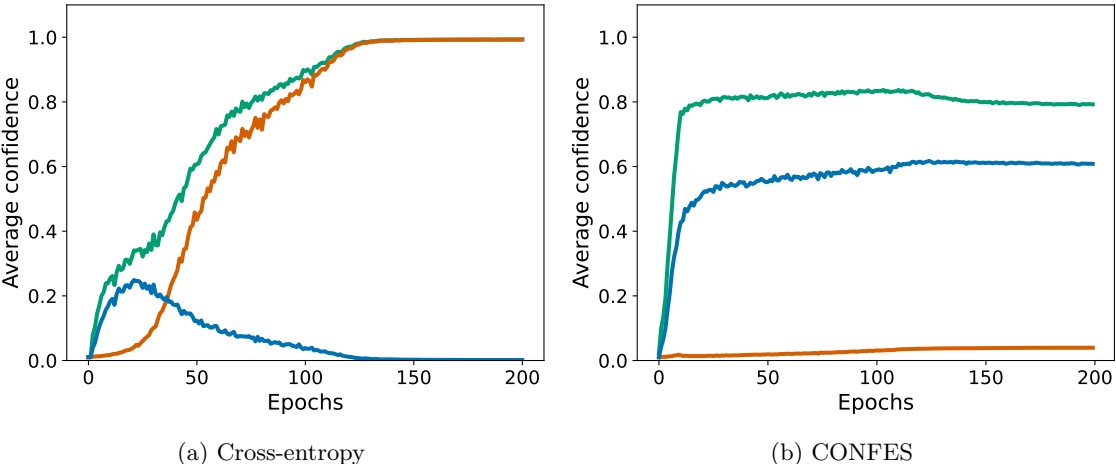

(a) Cross-entropy

(b) CONFES

Figure 3: **Effectiveness of confidence error based sieving**: Using naive cross-entropy training (left), the model confidence over noisy labels increases as the training moves forward. It implies that model is misled by the noisy samples. Confidence-based sieving (right), however, differentiates the noisy samples from the clean ones, and keeps the model confidence over the noisy labels as low as possible.

created by concatenating ($\oplus$) only the identified clean samples and their augmentations such that this dataset becomes as large as the initial training set. Finally, the model is trained on the augmented dataset, which consists of the most probably clean samples according to their confidence error.

**Why model confidence?** Prior works empirically and theoretically showed the effective role of model confidence in label noise settings Cheng et al. (2020); Zheng et al. (2020). The study by Cheng et al. (2020) adds a confidence regularization term to the loss function, which encourages confident (high probability) predictions and makes the elimination of noisy samples more efficient. The work from Zheng et al. (2020) theoretically proved that if the model confidence on a given sample is low, the probability of being noisy for that sample is high and bounded to an upper-bound which is dependent on how well the model has been trained. Given that, the study proposed a training algorithm called likelihood ratio test (LRT), which trains the model on all samples for a certain number of warm-up epochs (i.e. 30) to get an enough-trained model. Then, it uses the ratio of the confidence values on the original and predicted labels as a metric to separate the clean and noisy samples after the warm-up epochs. A sample for which the aforementioned ratio is close to 1.0 is considered as a clean sample. Motivated by these previous works, we incorporate the model confidence as the cornerstone of our proposed metric and sieving algorithm.

**Why confidence error?** The proposed confidence error metric has at least two advantages over likelihood ratio (a metric also based on confidence): (1) Confidence error enables the algorithm to start performing the sample sieving in the early epochs of training. Using the sieving threshold $\alpha_i$, the algorithm only incorporates the samples with confidence error less than $\alpha_i$ in the training instead of all samples. Applying similar threshold to likelihood ratio in warm-up epochs delivers much lower accuracy than using all samples based on our observations. (2) The confidence error is a more efficient metric than likelihood ratio for differentiating the clean samples from the noisy ones according to our experimental results provided in Figure 9 (at the Appendix). The results in the next section also verify this observation.

**Why sieving based on confidence error?** We use our previous experimental setup and train the model with the naive cross-entropy method and CONFES algorithm to answer this question. Figures 3a and 3b show the model confidence for the noisy, clean, and predicted labels (averaged over the corresponding samples) with cross-entropy and CONFES, respectively. According to Figure 3a, the confidence over noisy labels is very low at the early stages of cross-entropy training. However, as the training proceeds, the model confidence over noisy labels increases. At the end of training, the model confidence over predicted and noisy labels is close to each other. This indicates that the model has been misled by the noisy samples, wrongly considering

them as the true labels of the samples. This observation shows the importance of keeping the confidence over noisy labels as low as possible. Confidence-error based sieving, on the other hand, relies on the model confidence to separate the clean samples from noisy ones. This way, it makes the model to have very low confidence on the noisy samples throughout all training stages according to Figure 3b, leading to more robust model training against label noise.

## 4    Experiments

In this section, we draw a performance comparison between CONFES and recent baseline approaches Han et al. (2018); Liu et al. (2020); Chen et al. (2021a); Cheng et al. (2020); Bai et al. (2021) on three label noise settings: symmetric, pairflip, and instance-dependent. We first describe our experimental setting, and then provide and discuss the empirical results.

**Datasets**   We employ the CIFAR-10/100 datasets Krizhevsky et al. (2009) and Clothing1M datasets Xiao et al. (2015), which is naturally noisy and utilised as a standard benchmark for label noise training tasks. CIFAR-10/100 contain 50000 training samples and 10000 testing samples of shape $32 \times 32$ from 10/100 classes. For the CIFAR datasets, we perturb the training labels using symetric, pairflip, and instance-dependent label noise introduced in Xia et al. (2020), but keep the test set clean. For data augmentation/preprocessing, the training samples are horizontally flipped with probability 0.5, randomly cropped with size $32 \times 32$ and padding $4 \times 4$, and normalized using the mean and standard deviation of the dataset. Clothing1M is a real-world dataset of 1 million images of size $224 \times 224$ with noisy labels (whose estimated noise level is approximately 38% Wei et al. (2022); Song et al. (2019b)) and 10k clean test images in 14 classes. The data augmentation methods performed on this dataset include $256 \times 256$ resizing, $224 \times 224$ random crops and random horizontal flips. In the clothing1M training dataset, the number of samples for each class is imbalanced. Thus, we follow Li et al. (2020) and sample a class-balanced subset of the training dataset at each epoch.

**State-of-the-art methods**   On all of the examined datasets, we compare CONFES with the most recent related studies including: (1) standard cross-entropy loss (CE), (2) co-teaching Han et al. (2018) that cross-trains two models and uses the small-loss trick for selecting clean samples and exchanges them between the two models, (3) ELR Liu et al. (2020), an early-learning regularization method that leverages the model's output during the early-learning phase, (4) CORES$^2$ Cheng et al. (2020), a sample sieving approach that uses confidence regularization which leads the model towards having more confident predictions, (5) PES Bai et al. (2021), a progressive early-stopping strategy and (6) SLN Chen et al. (2021a) that improves regularization by introducing stochastic label noise. Co-teaching and CORES$^2$ are based on sample selection, ELR and SLN are regularization-based methods. For all of these methods, the specific hyper-parameters are set according to the corresponding manuscript or the published source code, if available.

**Neural Networks and optimization**   We conduct the experiments on a single GPU system equipped with an NVIDIA RTX A6000 graphic processor and 48GB of GPU memory. Our method is implemented in PyTorch v1.9. For all methods, we evaluate the average test accuracy on the last 5 epochs and for co-teaching, we report the average of this metric for the two networks. Following previous works  Li et al. (2020); Bai et al. (2021) on CIFAR-10 and CIFAR-100, we train the PreActResNet-18 He et al. (2016) model using the SGD optimizer with a momentum of 0.9, weight decay of 5e-4 and batch size of 128. The initial learning rate is set to 0.02 and is decreased by 0.01 in 300 epochs using a cosine annealing scheduler Loshchilov & Hutter (2017). On Cloting1M dataset, we adopt the setting from Li et al. (2020) and train the ResNet-50 model pre-trained on ImageNet Krizhevsky et al. (2012). The network is trained using SGD with momentum of 0.9 and weight decay of 1e-3 for 80 epochs, starting with learning rate of 0.002 which decreased by 10 at epoch 40. At each epoch the network is trained on 1000 mini-batches of size 32.

### 4.1    Evaluation

**CIFAR-10/100 datasets**   Tables 1 and 2 list test accuracy values for different noise types and noise rates on CIFAR-10 and CIFAR-100 datasets respectively. According to these tables, CONFES outperforms the competitors for all considered symmetric, pairflip, and instance-dependent noise types. Similarly, CONFES

Table 1: Test accuracy on CIFAR-10 for different noise types with noise level 40%.

| Method | Symmetric | Pairflip | Instance |
|---|---|---|---|
| CE | 66.61 ±0.4 | 59.25 ±0.1 | 66.04 ±0.2 |
| Co-teaching Han et al. (2018) | 87.42 ±0.2 | 84.57 ±0.2 | 86.90±0.1 |
| ELR Liu et al. (2020) | 85.74 ±0.2 | **86.15** ±0.1 | 85.37 ±0.3 |
| CORES$^2$ Cheng et al. (2020) | 83.9 ±0.4 | 58.38 ±0.6 | 76.71 ±0.4 |
| LRT Zheng et al. (2020) | 85.47 ±0.3 | 59.25 ±0.3 | 80.53 ±0.9 |
| PTD Xia et al. (2020) | 72.05 ±0.9 | 58.34 ±0.8 | 65.97 ±0.9 |
| PES Bai et al. (2021) | 90.55 ±0.1 | 85.56 ±0.1 | 85.63 ±0.5 |
| SLN Chen et al. (2021a) | 83.69 ±0.2 | 85.26 ±0.5 | 67.71 ±0.4 |
| CONFES (ours) | **90.62**±0.2 | **86.18**±0.3 | **90.28**±0.2 |

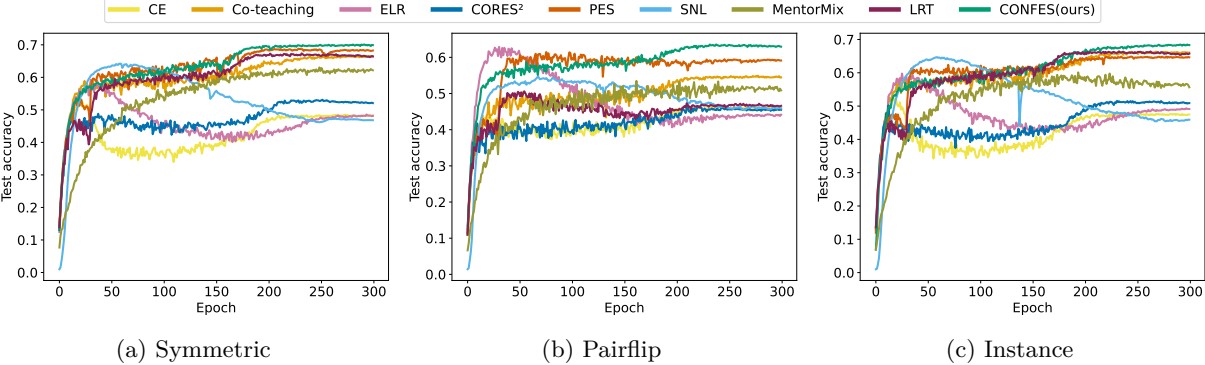

(a) Symmetric      (b) Pairflip      (c) Instance

Figure 4: Test accuracy on CIFAR-100 using PreAct-ResNet18: CONFES is robust against overfitting, whereas some of its competitors including SLN and ELR start overfitting after some epochs of training; noise level is 40%.

delivers higher accuracy than the competitors for different noise rates. Moreover, as the noise level increases, the accuracy gap between CONFES and its competitors widens in favor of CONFES. Figure 4 illustrates the test accuracy versus epoch for the different learning algorithms. As shown in the figure, CONFES is robust against overfitting because the corresponding test accuracy continues to increase as training moves forward, and stays at the maximum after the model converges. Some of the other algorithms such as SLN and ELR on the other hand suffer from an overfitting problem, where their final accuracy values are lower than the maximum accuracy they achieve.

Table 4 shows the accuracy values of CoTeaching, JoCor, and DivideMix if confidence error is used as the discriminator metric instead of the training loss. As shown in the table, the accuracy from these algorithms is enhanced by 2-5%, indicating that confidence error is not only effective as the main building block of the proposed CONFES algorithm, but also combined with other state-of-the-art methods such as Co-teaching, JoCor and DivideMix, which is a complex method employing data augmentation and relabeling. Furthermore, combining confidence error with other state-of-the-art methods (such as DivideMix) that guess or refine the noisy labels rather than excluding them, helps utilizing the noisy samples and learning their feature information as well.

**Clothing1M dataset** Table 3 summarises the performance of methods on Clothing1M dataset. CORES$^2$ and PES provide slight or no accuracy gain compared to the baseline cross-entropy training, respectively. CONFES, on the other hand, outperforms the competitors including ELR and SLN.

Table 2: Test accuracy on CIFAR-100 for various label noise types with different noise rates.

(a) **Symmetric**

| Method | 20% | 40% | 60% |
|---|---|---|---|
| CE | 63.46 ±0.7 | 47.85 ±0.4 | 29.59 ±0.3 |
| Co-teaching Han et al. (2018) | 71.54 ±0.3 | 66.26 ±0.1 | 58.82 ±0.1 |
| ELR Liu et al. (2020) | 63.59 ±0.1 | 48.33 ±0.2 | 30.37 ±0.1 |
| CORES$^2$ Cheng et al. (2020) | 65.99 ±0.5 | 52.26 ±0.2 | 34.61 ±0.2 |
| LRT Zheng et al. (2020) | 73.72 ±0.1 | 66.52 ±0.2 | 50.86 ±0.4 |
| MentorMix Jiang et al. (2020) | 71.52 ±0.2 | 61.96 ±0.2 | 44.38 ±0.3 |
| PES Bai et al. (2021) | 71.42 ±0.2 | 68.37 ±0.2 | 60.38 ±0.1 |
| SLN Chen et al. (2021a) | 60.48 ±0.1 | 46.98 ±0.2 | 28.50 ±0.2 |
| CONFES (ours) | **73.89**±0.1 | **69.63**±0.2 | **60.65**±0.1 |

(b) **Instance-dependant**

| Method | 20% | 40% | 60% |
|---|---|---|---|
| CE | 63.16 ±0.1 | 48.92 ±0.3 | 30.65 ±0.4 |
| Co-teaching Han et al. (2018) | 71.12 ±0.3 | 66.55 ±0.3 | 57.18 ±0.2 |
| ELR Liu et al. (2020) | 63.10 ±0.2 | 49.15 ±0.2 | 29.88 ±0.6 |
| CORES$^2$ Cheng et al. (2020) | 64.55 ±0.1 | 50.98 ±0.2 | 33.93 ±0.5 |
| LRT Zheng et al. (2020) | 73.14 ±0.2 | 65.32 ±0.6 | 45.37 ±0.1 |
| MentorMix Jiang et al. (2020) | 69.41 ±0.2 | 56.41 ±0.1 | 34.61 ±0.1 |
| PES Bai et al. (2021) | 71.65 ±0.3 | 64.83 ±0.2 | 41.10 ±0.5 |
| SLN Chen et al. (2021a) | 60.08 ±0.1 | 46.08 ±0.3 | 29.77 ±0.4 |
| CONFES (ours) | **73.59**±0.2 | **69.68**±0.2 | **59.48**±0.1 |

(c) **Pairflip**

| Method | 20% | 30% | 40% |
|---|---|---|---|
| CE | 64.31 ±0.3 | 55.77±0.1 | 45.62 ±0.4 |
| Co-teaching Han et al. (2018) | 69.59 ±0.2 | 64.04 ±0.4 | 55.42 ±0.5 |
| ELR Liu et al. (2020) | 62.05 ±0.5 | 54.44 ±0.2 | 44.31 ±0.3 |
| CORES$^2$ Cheng et al. (2020) | 63.85 ±0.2 | 54.88 ±0.3 | 45.34±0.2 |
| LRT Zheng et al. (2020) | 71.70 ±0.1 | 60.78 ±0.1 | 46.24 ±0.2 |
| MentorMix Jiang et al. (2020) | 69.65 ±0.1 | 62.01 ±0.1 | 50.97 ±0.2 |
| PES Bai et al. (2021) | 71.73 ±0.4 | 68.28 ±0.3 | 59.18 ±0.2 |
| SLN Chen et al. (2021a) | 61.82 ±0.3 | 53.67 ±0.2 | 45.72 ±0.2 |
| CONFES (ours) | **73.12**±0.1 | **71.34**±0.2 | **62.37**±0.4 |

Table 3: Test accuracy on Clothing1M dataset

| Method | CE | ELR | CORES$^2$ | PES | SLN | CONFES (ours) |
|---|---|---|---|---|---|---|
| Test Accuracy | 69.21% | 71.39% | 69.50% | 69.18% | 72.80% | **73.24%** |

## 5 Discussion

According to the experimental results, CONFES outperforms all baseline methods in the considered symmetric, pairflip, and instant-dependent noise settings. As the noise rate increases, the efficiency of the CONFES algorithm becomes more apparent (e.g. noise rate of 50% in CIFAR-100). Moreover, CONFES is robust to overfitting unlike some of its competitors such as SLN and ELR. This indicates that the underlying confidence

Table 4: Test accuracy for CONFES (CNF) combined with other approaches; dataset: CIFAR-100; model: PreActResNet18; noise rate: 40%.

| Method | Symmetric | Pairflip | Instance |
|---|---|---|---|
| Co-teaching Han et al. (2018) | 66.26 ±0.1 | 55.42 ±0.5 | 66.55±0.3 |
| CNF-Co-teaching | 69.94±0.1 | 57.90±0.2 | 69.51±0.1 |
| **Improvement** | **+3.68** | **+2.48** | **+2.96** |
| DivideMix Li et al. (2020) | 74.63±0.2 | 74.9±0.1 | 66.79±0.3 |
| CNF-DivideMix | 76.31±0.2 | 76.51±0.1 | 69.03±0.1 |
| **Improvement** | **+1.68** | **+1.61** | **+2.24** |
| JoCoR Wei et al. (2020) | 67.05±0.2 | 54.96±0.3 | 67.46±0.2 |
| CNF-JoCoR | 70.48±0.2 | 59.61±0.1 | 70.24±0.4 |
| **Improvement** | **+3.43** | **+4.65** | **+2.78** |

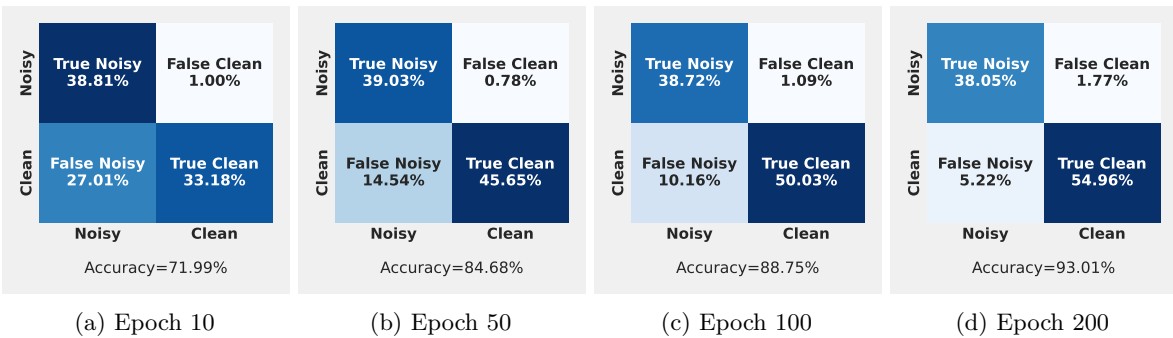

(a) Epoch 10          (b) Epoch 50          (c) Epoch 100          (d) Epoch 200

Figure 5: Confusion matrix for the CONFES algorithm. In early epochs, CONFES correctly identifies the majority of noisy labels (around 38% out of 40%), but wrongly identifies many clean labels as noisy ones (about 27%). As training proceeds, the algorithm not only still remains effective in identifying the noisy labels (around 38% out of 40%), but also correctly recognizes the clean labels (about 55% out of 60%). In the experiment, the dataset is CIFAR-100 with instance-dependant label noise of rate 40% trained using PreActResNet18.

error metric can effectively differentiate the clean labels from the noisy ones, and eliminate their adverse impact by excluding them from the training process.

Figure 5 shows the confusion matrix for the CONFES algorithm. According to the figure, CONFES is effective in recognizing the noisy samples from the beginning to the end of training, where it correctly identifies around 38% out of 40% of noisy labels. On the other hand, the algorithm wrongly identifies many clean samples as noisy in the early epochs (around 27%). However, as training moves forward, CONFES becomes more and more efficient in distinguishing the clean samples, where it correctly identifies around 55% out of 60% of the clean samples.

The initial sieving threshold $\alpha$ and number of warm-up epochs $T_w$ are the hyper-parameter of our proposed method. The per-epoch sieving threshold is computed using the aforementioned hyper-parameters. For CIFAR-100, we set $\alpha$=0.2 and $T_w$=30 for all noise types and noise rates. For CIFAR-10, $\alpha$=0.1 and $T_w$=25 in symmetric and instance-dependent noise types. For Clothing1M, $\alpha$ and $T_w$ are set to 0.05 and 3, respectively. In general, the value of these hyper-parameters should be set to higher values for more complex training processes, which depends on the model and dataset combination.

**Sensitivity analysis of hyper-parameters:** We investigate the sensitivity of CONFES to its hyper-parameters, i.e. the number of warm-up epochs ($T_w$) and the initial sieving threshold ($\alpha$), using the CIFAR-100 dataset with noise rate of 40% for symmetric, instance-dependent, and pairflip noise settings. To analyze

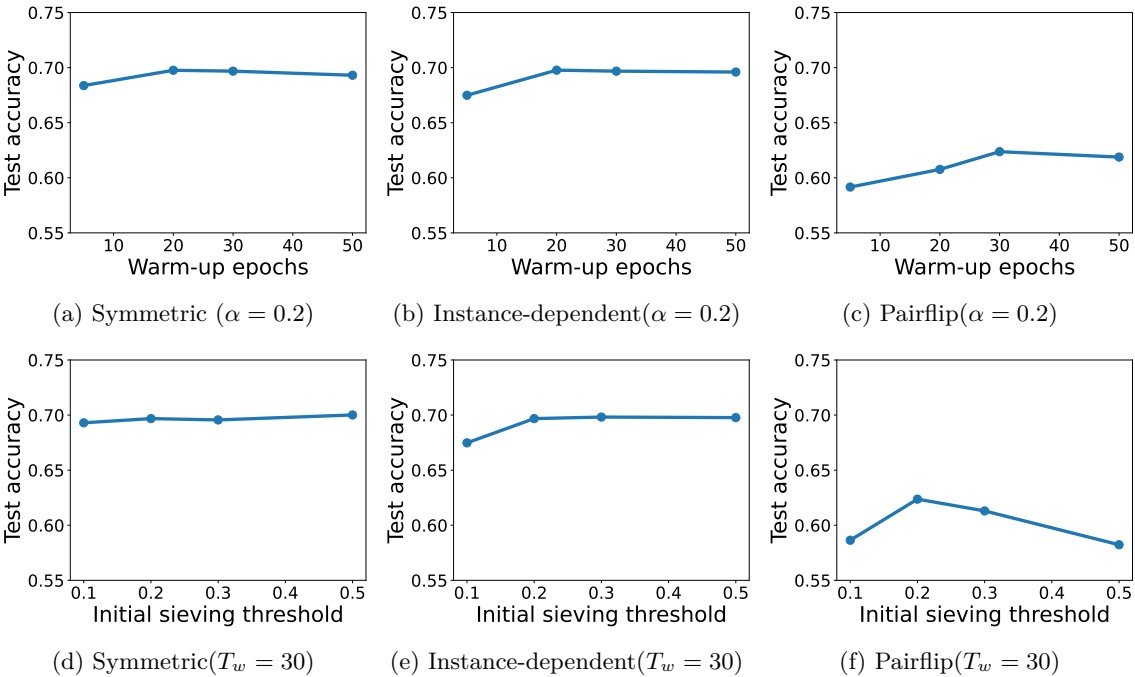

Figure 6: Sensitivity analysis of CONFES hyper-parameters $T_w$(number of warm-up epochs) and $\alpha$ (initial sieving threshold) for different noise types. The analyses are performed on CIFAR-100 with noise rate 40%.

the sensitivity to $T_w$, we set $\alpha = 0.2$ and use four different values for warm-up epochs: $T_w \in \{5, 20, 30, 50\}$. Similarly, we set $T_w = 30$ and try four different values for sieving threshold: $\alpha \in \{0.1, 0.2, 0.3, 0.5\}$. As shown in Figure 6, the accuracy reductions using the suboptimal hyper-parameter values compared to the optimal setting ($\alpha = 0.2$ and $T_w = 30$) are 1.6%, 2.3% and 4.1% for symmetric, instance-dependent, and pairflip noise settings, respectively, at the worst case. That is, CONFES is relatively robust against hyper-parameter value choices, making it easy to employ or tune by the practitioners.

**CONFES vs. baseline methods:** The CONFES strategy sieves the clean samples effectively and improves the robustness without adding too much overhead to the standard Cross-Entropy approach. In terms of computational overhead, CONFES has one additional forward pass for constructing the refined dataset, which only includes clean samples according to the confidence error metric. However, methods such as Co-teaching Han et al. (2018) employ two networks in the training process, which makes them substantially less computationally efficient compared to our approach. Although some methods such as PES Bai et al. (2021) perform well in the presence of symmetric label noise, their accuracy decreases in more complex noise settings such as instance-dependent, which is not the case for CONFES. Moreover, the accuracy of some other baseline methods such as LRT Zheng et al. (2020) and MentorMix Jiang et al. (2020) drastically reduces in highly noisy setting (e.g. with 60% noise rate). Approaches such as ELR Liu et al. (2020) and PES Bai et al. (2021) work well for datasets such as CIFAR-10, which are easy to classify, but their efficiency reduces on more challenging CIFAR-100 (or Clothing1M) dataset. CONFES, on the other hand, outperforms the compared baselines in different noise types (symmetric, instance-dependent and pairflip), with various noise levels (i.e. 20%, 40% and 60%), and on CIFAR-10, CIFAR-100 and Clothing1M datasets.

## 6 Conclusion

We present an effective label noise learning algorithm called CONFES for different noise settings such as symmetric, pairflip, and instance-dependent. CONFES is based on the proposed confidence error metric, which exploits the model's predictive confidence for discriminating between clean and noisy samples. We illustrate the efficacy of the confidence error metric in differentiation of the clean samples from the noisy ones. CONFES refines the training samples by keeping only the identified clean samples and filtering out

the noisy ones. Our empirical results verify the robustness of CONFES under different noise type scenarios, especially when noise levels are high. Moreover, we demonstrate that confidence error can be employed by other algorithms such as co-teaching and DivideMix to further improve the model accuracy.

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

## A  Appendix

**More experiments on the effectiveness of confidence error** : We extended the experiments corresponding to Fig.1 and Fig.2 of the main manuscript, to the symmetric and pairflip label noise. Experiments are conducted using PreAct-ResNet18 and CIFAR-100 with noise level of 40%.

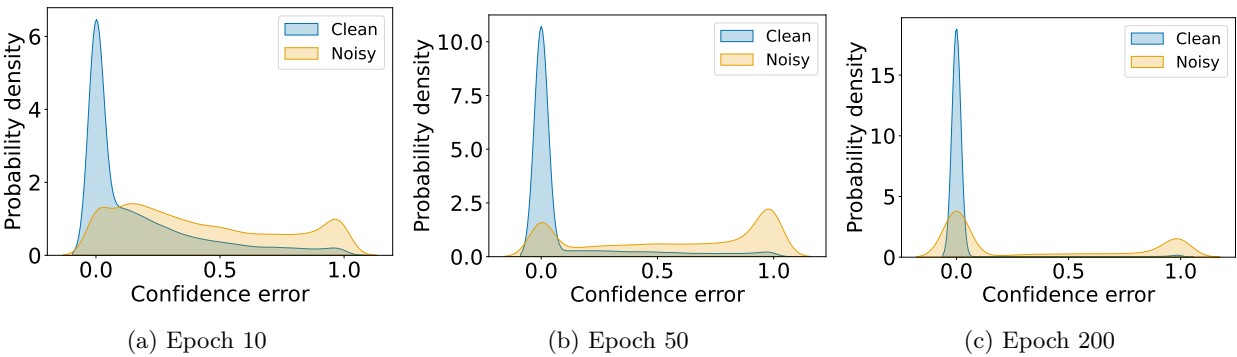

(a) Epoch 10        (b) Epoch 50        (c) Epoch 200

Figure 7: Distributions of **confidence error** values for **pairflip** label noise

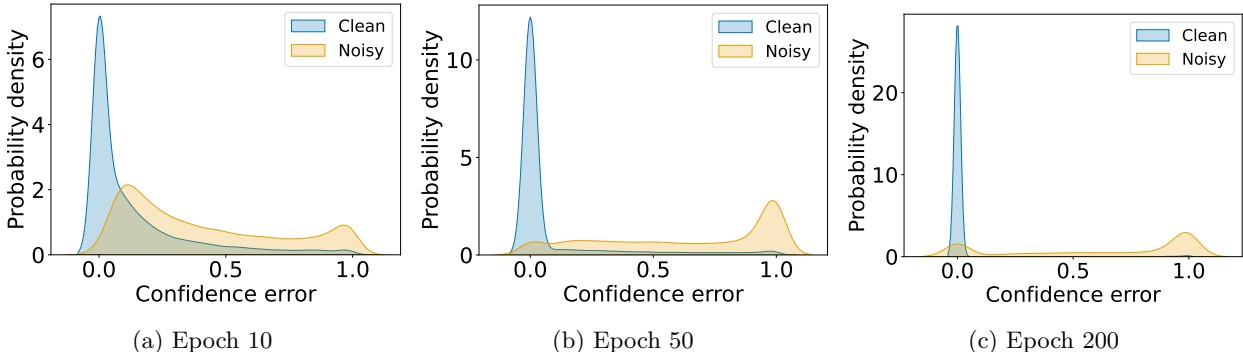

(a) Epoch 10        (b) Epoch 50        (c) Epoch 200

Figure 8: Distributions of **confidence error** values for **symmetric** label noise

**More details on experiment setup** For all experiments on the CIFAR-10, CIFAR-100, and Clothing1M datasets, there are some general hyper-parameters such as learning rate, batch size and weight decay, specified in the original manuscript and are summarized in Table 5. The source code for CONFES is included in the supplementary material and will be publicly available. The method-specific hyper-parameters used in the experiments are set based on the corresponding manuscript or the published source code: Co-teaching Han et al. (2018)[1], ELR Liu et al. (2020) [2], CORES[2] Cheng et al. (2020)[3], PES Bai et al. (2021)[4], SLN Chen et al. (2021a) [5], DivideMix Li et al. (2020)[6], JoCoR Wei et al. (2020) [7], LRT Zheng et al. (2020)[8], MentorMix Jiang et al. (2020)[9] and PTD Xia et al. (2020)[10].

---

[1] https://github.com/bhanML/Co-teaching

[2] https://github.com/shengliu66/ELR

[3] https://github.com/UCSC-REAL/cores

[4] https://github.com/tmllab/PES

[5] https://github.com/chenpf1025/SLN

[6] https://github.com/LiJunnan1992/DivideMix

[7] https://github.com/hongxin001/JoCoR

[8] https://github.com/pingqingsheng/LRT

[9] https://github.com/LJY-HY/MentorMix_pytorch

[10] https://github.com/xiaoboxia/Part-dependent-label-noise

Table 5: General training hyperparameters (common for all methods of comparison)

|                    | CIFAR-10         | CIFAR-100        | Clothing1M           |
|--------------------|------------------|------------------|----------------------|
| model              | PreAct-ResNet18  | PreAct-ResNet18  | Pre-trained ResNet50 |
| batch size         | 128              | 128              | 32                   |
| learning rate (lr) | 2e-2             | 2e-2             | 2e-3                 |
| lr decay           | Cosine annealing | Cosine annealing | By 0.1 at 40th       |
| weight decay       | 5e-4             | 5e-4             | 1e-3                 |
| epochs             | 300              | 300              | 80                   |

**Comparison between CONFES and LRT Zheng et al. (2020)** we employed the experiment designed for Fig.1 and Fig.2 of the main manuscript, to examine how-well CONFES can seprate the noisy labels from the clean labels compare to LRT.

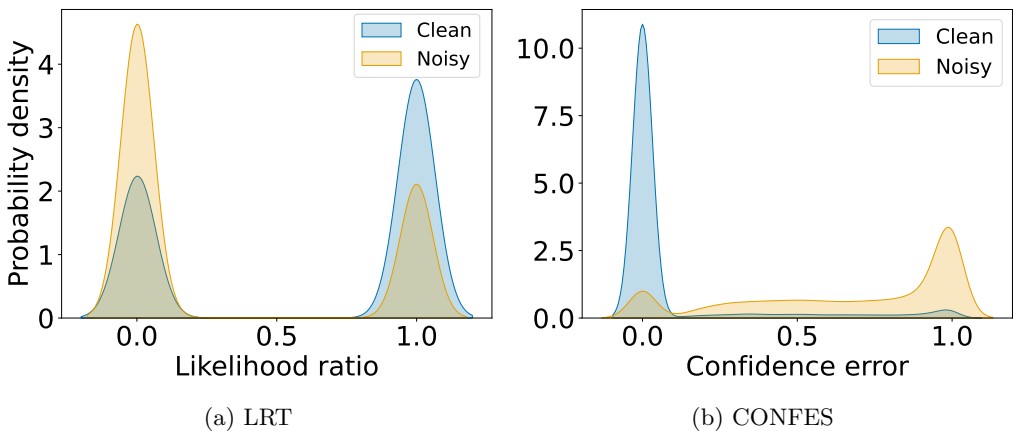

(a) LRT

(b) CONFES

Figure 9: Distributions of **likelihood ratio** (employed in LRT) and **confidence error** (employed in CONFES) values for 60% instance-dependent label noise at epoch 200. The distributions of confidence error for noisy and clean samples are more dissimilar than that of likelihood ratio, indicating that confidence error is a more effective metric than likelihood ratio for sieving the samples.

**Instance-dependent label noise:** In order to generate the instance-dependent label noise in the experiments, we followed the previous works Cheng et al. (2020); Yao et al. (2020); Bai et al. (2021); Chen et al. (2021a) and employed the following algorithm proposed in Xia et al. (2020):

---

**Algorithm 2:** Instance-dependent Label Noise Generation

---

**Input:** Clean samples $\{(x_i, y_i)\}_{i=1}^n$, Noise rate $\tau$
**Output:** Noisy samples $\{(x_i, \tilde{y}_i)\}_{i=1}^n$

**1** Sample instance flip rates $q \in \mathbb{R}^N$ from the truncated normal distribution $\mathcal{N}(\tau, 0.1^2, [0, 1])$
**2** Independently samples $w_1, ..., w_c$ from the standard normal distribution $\mathcal{N}(0, 1^2)$
**3 for** $i = 0, ..., n$ **do**
**4** $\quad$ $p = x_i \times w_{y_i}$ /* Generate instance dependent flip rate $\quad\quad\quad$ */
**5** $\quad$ $p_{y_i} = -\infty$ /* control the diagonal entry of the instance-dependent transition matrix $\quad$ */
**6** $\quad$ $p = q_i \times softmax(p)$ /* make the sum of the off-diagonal entries of the $y_i$-th row to be $q_i$ $\quad$ */
**7** $\quad$ $p_{y_i} = 1 - q_i$ /* set the diagonal entry to be $1 - q_i$ $\quad\quad\quad$ */
**8** Randomly choose a label from the label space according to the possibilities p as noisy label $y_i$
**9 return** Noisy samples $\{(x_i, \tilde{y}_i)\}_{i=1}^n$

---

