# OpenReview forum: "Label Noise-Robust Learning using a Confidence-Based Sieving Strategy"
_TMLR — Rejected by TMLR_

### Review · Reviewer_jcnm · 2023-01-03

**Summary Of Contributions:**

This paper proposes a new approach CONFES for noisy label learning. To be specific, they first experimentally demonstrate that distributions of loss values for clean and noisy labels are similar, and consequently, loss value is an inefficient metric to distinguish between the clean labels and noisy ones. Then, they propose a metric called confidence error, i.e., the difference between
the probability assigned to the predicted label and the probability associated with the original label, to discriminate between the clean and noisy labels. Finally, clean training samples are sieved during each epoch to build new datasets.


**Audience:**

Yes

**Claims And Evidence:**

Yes

**Requested Changes:**

1. Though the experimental results are nice, the core of the proposed method, i.e., the sieving metric ‘confidence error’ seems heuristic, which needs more discussions in Page 6, Paragraph ‘Why confidence error works effectively in label noise training’.
2. The core idea of another paper [1] is similar with this paper. [1] uses the division between the probability assigned to the predicted label and the probability associated with the original label while this paper uses subtraction. Also, [1] provides a theoretical insight on why such a mechanism works. The differences between [1] and this paper are suggested to be discussed.
3. Something is confusing in Figure 1 and Figure 2. The scale of Y-axis (Probability density 0-0.15) in Figure 1 is far away from that (Probability density 0-2) in Figure 2. The visualization should be explained more explicitly.
4.  In Figure 3, one example is not persuasive. It might be better to illustrate the whole noisy examples.

**Strengths And Weaknesses:**

Strength:
1. The paper proposes an alternative metric, which is different from the loss value used before and seems reasonable, to sieve training examples more effectively. Experimentally, the proposed method has an impressive performance.
2. The paper vividly visualizes the drawbacks the previous method using the loss value has.

Weakness:
1. Though the experimental results are nice, the core of the proposed method, i.e., the sieving metric ‘confidence error’ seems heuristic, which needs more discussions in Page 6, Paragraph ‘Why confidence error works effectively in label noise training’.
2. The core idea of another paper [1] is similar with this paper. [1] uses the division between the probability assigned to the predicted label and the probability associated with the original label while this paper uses subtraction. Also, [1] provides a theoretical insight on why such a mechanism works. The differences between [1] and this paper are suggested to be discussed.
3. Something is confusing in Figure 1 and Figure 2. The scale of Y-axis (Probability density 0-0.15) in Figure 1 is far away from that (Probability density 0-2) in Figure 2. The visualization should be explained more explicitly.
4.  In Figure 3, one example is not persuasive. It might be better to illustrate the whole noisy examples.
[1] Zheng, Songzhu, et al. "Error-bounded correction of noisy labels." International Conference on Machine Learning. PMLR, 2020.

---

> ### Author Response · Authors · 2023-01-24
> **Response to Reviewer jcnm**
>
> We thank the reviewer for the helpful comments, and kindly invite the reviewer to check out the revised draft too.
>
> 1. "Though the experimental results are nice, the core of the proposed method, i.e., the sieving metric ‘confidence error’ seems heuristic, which needs more discussions in Page 6, Paragraph “Why confidence error works effectively in label noise training.”
>
> We added more discussion on Page 6 and explained why we incorporated the model confidence in the first place (paragraph “why model confidence”), why the proposed confidence error works more effective than likelihood ratio, which is also based on model confidence (paragraph “why confidence error”), and why sieving based on confidence error makes the model more robust to label noise (paragraph “why sieving based on confidence error”).  In summary, we argue that the lower model confidence implies the higher probability of being noisy according to the previous studies. Moreover, sieving based on confidence error keeps the model confidence low on the noisy labels throughout training, leading to a more robust model.
>
> 2. "The core idea of another paper [1] is similar with this paper. [1] uses the division between the probability assigned to the predicted label and the probability associated with the original label while this paper uses subtraction. Also, [1] provides a theoretical insight on why such a mechanism works. The differences between [1] and this paper are suggested to be discussed."
>
> We included paper [1], likelihood ratio test (LRT) as one of our baseline methods and reported its performance on CIFAR100 (Table 2 on page 9, Figure 4 on page 8) and CIFAR10 (Table 1, page 8) datasets in presence of symmetric, instance-dependent and pairflip label noise with different noise rates (i.e. 20%, 30%, 40% and 60%). Furthermore, we discussed the difference between our proposed method and likelihood ratio in Page 6, paragraph “why confidence error?”. In addition to that, we designed an experiment to compare the effectiveness of our proposed metric and the likelihood ratio, the result of which has been visualized in Figure 9 (Appendix, paragraph “Comparison between CONFES and LRT”).  In summary, confidence error has at least two advantages over likelihood ratio: First, it enables the algorithm to start sieving the samples at early epochs very effectively, which is not the case for likelihood ratio. Second, confidence error is a better discriminative metric than likelihood ratio based on our experiments (Figure 9 in the appendix). Our accuracy results confirm that CONFES provides higher accuracy than LRT for all three considered noise settings and noise rates.
>
> 3. "Something is confusing in Figure 1 and Figure 2. The scale of Y-axis (Probability density 0-0.15) in Figure 1 is far away from that (Probability density 0-2) in Figure 2. The visualization should be explained more explicitly."
>
> The y-axis in Figure 1 and 2 represents the probability density function (PDF), which has been estimated using the Kernel Density Estimation (KDE) method. As the name implies, the PDF indicates the probability density but not probability. Thus, its value does not have to be in [0,1]. However, the integral over PDF, or the area under PDF curve, should be 1 (a basic probability theorem). Considering that the CE loss values have wider range than the confidence error values ( 0-30 vs.  0-1 in our case), the corresponding  probability density values have narrower range (0-0.15 vs. 0-2), which is consistent with the fact that the area under the PDF curve must be 1.
>
> 4. "In Figure 3, one example is not persuasive. It might be better to illustrate the whole noisy examples. "
>
> We updated Figure 3 such that instead of considering only one sample, we take the average over the confidence values from all corresponding samples. As shown in Figure 3, after the early epochs of CE training (about 20-30 epochs),  the model memorizes the noisy labels and becomes more and more confident in performing the wrong predictions. However, using CONFES, the model keeps the confidence on noisy labels as low as possible during the course of the training.

---

### Review · Reviewer_S2G3 · 2023-01-06

**Summary Of Contributions:**

The paper studies the problem of noisy label learning. The paper proposes a metric called confidence error to identify samples with potentially the noisy given label. The proposed algorithm CONFES removes the identified noisily-labeled samples and only trains the model on the clean samples. CONFES does such a process iteratively. The paper conducts some experiments to show that the proposed metric is more accurate and reliable than the loss metric. The paper shows improved empirical performance compared to some baselines.

**Audience:**

Yes

**Claims And Evidence:**

Yes

**Requested Changes:**

1. More empirical or theoretical studies about the properties of the proposed metric. For example, why the proposed metric is more reliable than the loss metric, and what's their relationship?

2. More comprehensive empirical studies, including sensitivity analysis of the hyper-parameters, results on more noise levels, and comparisons with stronger baselines.

**Strengths And Weaknesses:**

Strengths:
1. The proposed method is very simple and easy to implement.
2. The proposed method shows good empirical performance on certain tasks.


Weaknesses:
1. The paper proposes to use the confidence error metric. However, the understanding of such a metric is quite limited. There is no theoretical understanding, and the empirical experiments don't have strong implications. Particularly, the empirical experiments mostly show that the proposed metric is more reliable, but we don't understand why it is the case.

2. The proposed confidence error metric is related to the loss metric. Intuitively, a large confidence error should correspond to a large loss and vice versa. The paper does not discuss deeply the relationship between the two and why the proposed loss metric should work better. There is no hypothesis given to explain the different behaviors of the two metrics.

3. The proposed method only utilizes the identified clean labels and ignores the feature information in the noisy label samples. Could this method be applied together with other methods that utilize noisy samples?

4. The empirical studies could be stronger or more convincing. The noise level reported is centered around 40%. More results on a low noise level (e.g., 20%) or a high noise level (e.g., 80%) could make the results more convincing. Also, sensitivity studies are not presented for some of the hyper-parameters. For example, how does the number of warm-up epochs affect the performance?

5. Did not compare with some baselines with better performance, such as:
[1] Beyond Synthetic Noise: Deep Learning on Controlled Noisy Labels
[2] Robust Curriculum Learning: from clean label detection to noisy label self-correction


Minor:
1. Many references throughout the paper may have the wrong format (e.g., citep v.s. citet). I am not sure about the formatting requirements of the submission, though.
2. A sample is denoted as S; a lowercase letter may be more appropriate.

---

> ### Author Response · Authors · 2023-01-24
> **Response to Reviewer S2G3**
>
> We thank the reviewer for the constructive comments, and kindly invite the reviewer to check out the revised draft too.
>
> 1. "The empirical studies could be stronger or more convincing ..."
>
> We included noise rates 20% and 60% for symmetric and instance-dependent noise as well as noise rates 20% and 30% for pairflip. Based on our results, we do not anticipate that adding 80% noise would offer additional insight. Note that in pairflip noise every label might flip to its neighbor class (depending on noise rate), noise level of 40%-45% is considered as high in the literature. Specifically, based on our experiments if the noise rate is 50% or higher in pairflip, the methods will perform as bad as CE.  As shown in Table1, CONFES outperforms the baseline methods in all the considered noise settings.
>
> 2. "Sensitivity studies are not presented for some ..."
>
> We included the sensitivity analysis for the hyper-parameters of CONFES, i.e. number of warm-up epochs and initial sieving threshold on CIFAR-100 with noise level 40% (Figure 6). For instance-dependent and symmetric noise, test accuracy decreases about 2% in the worst case.  For pairflip, our method is more sensitive to the initial sieving threshold than the number of warm-up epochs, but in the worst case it changes less than 4%. So, CONFES is relatively robust to hyperparameter choices.
>
> 3. "The paper proposes to use the confidence error ..."
>
> We added further discussion on Page 6 and explained  “why model confidence”,  “why confidence error”) and “why sieving based on confidence error”). In summary, it has already been shown in previous works that model confidence can play an effective role in label noise. In particular, a previous study theoretically proves high model confidence implies low probability of being noisy given an enough-trained model. So, we incorporated model confidence as the cornerstone of our proposed confidence error metric. Confidence error can start sieving the samples efficiently from the beginning of the training instead of using all samples for a particular number of warm-up epochs (enough-trained model assumption is alleviated for confidence error). This is not the case for likelihood ratio (also based on model confidence), which requires an enough-trained model to perform effective sieving. Moreover, confidence error is a better discriminative metric compared to likelihood ratio according to our experiments (Figure 9, appendix).
>
> 4. "The proposed confidence error metric is related to the loss ..."
>
> We discussed in the second paragraph of page 6, that loss value by itself is not necessarily an effective metric for  discriminating between the clean and noisy samples. Previous works have proposed techniques such as loss correction or co-training to increase the robustness of cross-entropy loss to label noise. However, model confidence is more reliable than loss. In Figures 1 and 2, we visualized the different behavior of loss (as a sieving metric) and confidence error. The distributions of the loss values for the clean and noisy labels are relatively similar (Figure 1) while the distributions of confidence error values for the clean and noisy labels are very dissimilar.  Moreover, in Table 4, we picked some popular methods that use loss as a discriminative metric and replaced it with confidence error. The accuracy of these algorithms is enhanced by 2-5%, indicating that confidence error is not only effective as the main building block of CONFES, but also combined with other SOTA methods.
>
> 5."The proposed method only utilizes the identified clean labels ..."
>
> Yes, the confidence error can be used along with other existing methods as shown in Table 4 (page 10). DivideMix, in particular, utilizes noisy samples by performing guessing or refining the noisy labels such that they can learn from the noisy samples too. The results show that confidence error can improve the  already existing methods including DivideMix as discussed in the 2nd paragraph of page 8.
>
> 6. "Compare with other baselines such as  [1] ..."
>
> We included [1], as one of our baseline methods and reported its performance on CIFAR-100 with different noise types and rates in Table 2 (MentorMix). CONFES outperforms this approach in all the aforementioned settings. The other work [2], is unfortunately not open-source and we could not include it to our comparisons. Following the suggestion of the reviewer jcnm, we also evaluated the performance of a new work (Zheng, et al., 2020) on CIFAR-10/100 with different noise types and rates (Table 2, LRT).  Addressing the comment from reviewer xj1Z, we also included (Xia et al. 2020)(Table 1, PTD) to our experiments on CIFAR-10 with various noise settings. As shown in Table 1, Table 2 and Figure 4, CONFES outperforms all of these newly added baselines too.
>
> 7. "Many references throughout the paper ..."
>
> We used the TMLR template and the references are in TMLR bibliography style.
>
> 8. "A sample is denoted as S ..."
>
> We changed ‘S’ to ‘s’.

---

### Review · Reviewer_xj1Z · 2023-01-10

**Summary Of Contributions:**

This paper proposes a new metric to differentiate noisy labels and clean labels. The paper shows empirically the effectiveness of the metric and proposes a method that filters out the noisy examples in the early phase of learning. Experimental results validate the effectiveness of the proposed method.

**Audience:**

Yes

**Claims And Evidence:**

No

**Requested Changes:**

The paper has tried to provide some reasoning on why the metric is effective on Page 6. However, from the description, I cannot see why the proposed metric works better than the small-loss trick used widely in previous works. Following the same story provided on Page 6, if any instance which belongs to "Cat" is mistakenly labeled as "Horse", it will also incur a large loss in the early phase of learning, and the large loss could help exclude these noise samples easily. Better reasoning or intuition is required to justify or motivate why confidence error works better than the small loss trick in label noise training.

Since I did not see a significant difference between the proposed metric and the small loss metric, I am considering why the proposed method works better as shown in Section 4. I noticed that the paper uses data augmentation on Line 7 of the algorithm. However, I do not think existing methods have used data augmentation for a fair comparison. At least the original paper on these methods may not use data augmentation but it still, works fine. The paper needs a more fair comparison with baselines in the aspect of using data augmentation or not.

I am also curious about how the paper generates the instance-dependent label noise. The [Xia et al. 2020] cited focuses on a special instance-dependent label noise, but may not be a fully instance-dependent one. Moreover, although the paper uses [Xia et al. 2020] as the data generation method, it does not compare with this method at all. Can I ask why not compare with these methods since employing the same data generation process?

Regarding the motivation results in Figure 1, I noticed that it simply considers the first 40% of the samples with lower loss values as clean. I am wondering will this "treating as clean" method be too naive to represent existing methods. Existing methods such as Co-teaching has more advanced strategies on how to differentiate clean and noisy data. If such a strategy is wrongly used in the early phase of learning, it is natural that more noisy data are used in training resulting in a bad result as shown in Figure 1. More details on whether a fair strategy is used for sample selection should be provided.

I also have some minor comments on the presentation, which may be easily fixed in the revision phase:

1. After Eq. 4, the paper said that "the confidence error states the possibility that the predicted class is the same as the original class". However, this statement is not accurate at least. Eq. 4 only gives the difference between predicted confidence values but does not test whether \hat y equals \tidle y. The presentation here may need adjustment

2. Instance-dependent noise is first mentioned in paragraph 2 and could serve as one of the main contributed areas of the paper. However, in introducing related methods in Paragraph 3, Section 1, and Section 2, no special focus is given to instance-dependent noise and how to simulate the noise in experiments. The paper needs more content in this aspect.

3. Previous methods have already demonstrated their effectiveness in their experiments. I am wondering have the current paper beat them absolutely, or just in some of the noise settings. The paper may need to discuss under which setting, or situation the proposed method could work better than the baselines.


**Strengths And Weaknesses:**

Strengths,
1. The paper proposes a novel metric to differentiate clean labeled data and noisy labeled data, which has been demonstrated to be effective at differentiation empirically.
2. An empirical effective method is proposed based on the metric
3. The paper is well-written and well-organized. It is easy to follow.


Weaknesses,
Some more justification is required for the correctness and effectiveness of the proposed metric, and the proposed method.

---

> ### Author Response · Authors · 2023-01-24
> **Response to  Reviewer xj1Z**
>
> We thank the reviewer for the helpful comments, and kindly invite the reviewer to check out the revised draft too.
>
> 1. "The paper has tried to provide some reasoning on why...."
>
>  We discussed in the second paragraph of page 6, that loss value by itself is not necessarily an effective metric to  discriminate between the clean and noisy samples. As shown in Figure 1, the distribution of loss values are relatively similar compared to confidence error showed in Figure 2. Consequently, the previous loss-based works rely on techniques such as loss correction or co-training to make the loss metric more robust to label noise. Confidence error, however, is an efficient metric by itself. We discussed the reasons behind its efficiency on Page 6 and explained why we incorporated the model confidence (paragraph “why model confidence”), why the proposed confidence error works (paragraph “why confidence error”) and why sieving based on confidence error makes the model robust to label noise (paragraph “why sieving based on confidence error”). Furthermore, in Table 4, we picked some popular methods that use loss as a discriminative metric  and replaced it with confidence error. As a result, the accuracy of these algorithms is enhanced by 2-5%, indicating that confidence error is not only effective as the main building block of CONFES, but also combined with other state-of-the-art methods (page 8, paragraph 2).
>
> 2. "Since I did not see a significant difference between the proposed metric ..."
>
> The data augmentation that line 7 of the algorithm refers to is not generating any new samples. We only replace the possibly noisy samples with a duplication of the possibly clean samples such that the number of training samples in our algorithm is the same as that in other baseline methods (e.g. 50k for CIFAR). The only exception is co-teaching in which the identified noisy samples are excluded from the training (without being replaced) but instead, it uses an additional model. So, we think the setting for comparing baseline methods is fair in the aspect of data augmentation.
>
> 3. "I am also curious about how the paper generates the instance-dependent ..."
>
> We included [Xia et al.2020] to our baseline methods and evaluated its performance on CIFAR-10 dataset with different noise types and 40% noise rates in Table 1 (on page 8, PTD method). The performance of this approach is not good (65.97%, 72.05% and 58.34% for instance-dependent, symmetric and pairflip noise respectively). Note that the best accuracy reported in the original manuscript for 40% instance-dependent noise is 58.62% (other noise types not tested in the original manuscript).
>
> 4. "Regarding the motivation results in Figure 1, I noticed that ... "
>
> The sieving strategy employed in the experiments associated with Figure 1, is the same strategy as used in co-teaching and the only difference is that co-teaching employs an additional model (co-training), which is not the case here to have a more fair comparison. In co-teaching, every model is trained on the samples whose loss is the smallest (e.g. 40% of samples if noise rate is 60%) according to the other model. For figure 1, we also train the model on the samples with the smallest loss (40% of the samples) and exclude the rest without using a second model (for a fair comparison). In simple words, the aim of Figure 1 is to show the loss metric is not efficient by itself. There are methods such as co-teaching, which relies on an additional model to make the loss metric more efficient. Confidence error, on the other hand, is effective by itself, and does not need additional model or loss correction methods. Moreover, if it is used as a discriminative metric instead of loss in co-teaching, it can improve the accuracy by 2-3% (Table 4).
>
> 5. "After Eq. 4, the paper said that "the confidence error states ...."
>
> We adjusted this statement as follows: “the confidence error states how much the model confidence on the original class is far from the model confidence on the predicted class.”
>
> 6. "Previous methods have already demonstrated their  ..."
>
> We added a paragraph entitled “CONFES vs. baseline methods:” on page 11: In summary, some of the baseline methods perform well on simple noise types such as symmetric noise, slightly noisy settings (e.g. 20% noise), or simple datasets such as CIFAR-10. However, their performance drastically drops when the noise is more complex, the dataset is more complicated or the setting is highly noisy. On the other hand, CONFES, outperforms the compared baselines regardless of noise type and with various noise levels, and on synthetic and real-world datasets.
>
> 7. "Instance-dependent noise is first mentioned in paragraph 2 ..."
>
> Following the previous works, we generated the instance-dependent label noise using the method proposed in (Xia et al. 2020). To clarify, we explained the algorithm of generating instance-dependent noise in the appendix (paragraph “Instance-dependent label noise” on page 16).

---

### Decision · Action_Editors · 2023-03-07

**Recommendation:** Reject

**Comment:**

The submission gives an interesting first look into a potentially interesting rule for confidence filtering, but does not provide sufficient theoretical motivation to support the methods claimed generality.  A majority of reviewers indicated that it did not cross the threshold of support, and the one reviewer who leaned accept qualified their support indicating that the empirical results "may" be useful, and that "The weak side of the paper lies in explanations."

On the balance, this falls below the threshold of support that I would expect for this journal, and the claims are misaligned with the degree of support from the experiments.

The submission could benefit from rewriting to align claims with empirical evidence, eventually combined with additional theoretical motivation and/or empirical support.  In this case, it could be suitable for a major revision.

**Audience:**

The topic is suitable for a TMLR audience, and is clearly related to a common issue in machine learning.

**Claims And Evidence:**

The submission claims that a specific metric, confidence error, performs better than cross-entropy loss to determine mislabeled samples.  This is then employed in a framework for confidence based training, showing that confidence error performs better in this setting for CIFAR-10, CIFAR-100, and Clothing1M datasets.  From the paper:

>In summary, we make the following contributions:
>• We demonstrate that loss value by itself is not an effective metric to discriminate between the clean
and noisy samples.
>• To address this shortcoming, we introduce the confidence error as a novel alternative metric and
illustrate that it can efficiently differentiate clean samples from noisy ones.
>• We propose the CONFES learning algorithm, which leverages the confidence error as a core building
block to effectively sieve the training samples in an online fashion during training.
>• Through extensive experiments, we show that CONFES outperforms the state-of-the-art learning
algorithms of comparison, in terms of accuracy in symmetric, pairflip, and instance-dependent label
noise scenarios.
>• We combine CONFES with other learning algorithms including CoTeaching, JoCor, and DivideMix
and illustrate the combination provides further improved accuracy.

Evidence is on 3 datasets, and claims are made too strongly compared to the support for them.  For example, although evidence is limited (primarily CIFAR), statements are made as though they are fully general, e.g. "However, our observations (Fig 1) illustrate the distributions of the loss values for clean and noisy samples overlap widely, implying that a lot of noisy samples have small loss values and vice versa."  Fig 1 & 2 are entirely based on CIFAR with a single ResNet-18 model.

> Middle of p.5 "As shown in Figure 2.... Given that, we conclude that confidence error is a more efficient metric...."

The authors have not claimed that there are theoretical results supporting their conclusions, so the evidence is entirely empirical.  Given the experiments limitations in type of data and models and lack of theoretical support, the claims that the proposed heuristic is more efficient and general are not sufficiently supported by the results.